# Engineering micro oxygen factories to slow tumour progression via hyperoxic microenvironments

Weili Wang[1,6], Huizhen Zheng[1,6], Jun Jiang[1], Zhi Li[2], Dongpeng Jiang[3], Xiangru Shi[3], Hui Wang[1], Jie Jiang[1], Qianqian Xie[1], Meng Gao[1], Jianhong Chu [3], Xiaoming Cai[4], Tian Xia[5] & Ruibin Li [1]✉

While hypoxia promotes carcinogenesis, tumour aggressiveness, metastasis, and resistance to oncological treatments, the impacts of hyperoxia on tumours are rarely explored because providing a long-lasting oxygen supply in vivo is a major challenge. Herein, we construct micro oxygen factories, namely, photosynthesis microcapsules (PMCs), by encapsulation of acquired cyanobacteria and upconversion nanoparticles in alginate microcapsules. This system enables a long-lasting oxygen supply through the conversion of external radiation into red-wavelength emissions for photosynthesis in cyanobacteria. PMC treatment suppresses the NF-kB pathway, HIF-1α production and cancer cell proliferation. Hyperoxic microenvironment created by an in vivo PMC implant inhibits hepatocarcinoma growth and metastasis and has synergistic effects together with anti-PD-1 in breast cancer. The engineering oxygen factories offer potential for tumour biology studies in hyperoxic microenvironments and inspire the exploration of oncological treatments.

Hypoxia is the most pervasive characteristic of microenvironments of solid tumours[1,2] and arises from an imbalance between insufficient oxygen supply and increased oxygen consumption by rapidly proliferating cancer cells. Consequently, cancer cells resort to multiple adaptive pathways and genomic changes for survival in hypoxic environments[3]. The transcription factor hypoxia-inducible factor 1α (HIF-1α), the most recognized mediator of hypoxic responses, plays a central role in stimulating neovascularization in tumours to enhance oxygen and nutrient supply[4]. Paradoxically, these vessels are often irregularly organized (e.g., twisted, hyperpermeable and blind-ended structures) and have defects in oxygen diffusion or perfusion[5], resulting in expansions of hypoxic regions in tumours. Concomitantly, the hypoxic microenvironment, a hallmark of malignant tumours, has been reported to be not only the primary barrier shielding the

tumour from various therapies by creating an immunosuppression environment[6], activating the DNA repair pathway[7] and by enabling autophagic flux[8] but also a promoter of carcinogenesis[9], tumour invasiveness and metastasis[1,2]. These findings inspired the exploration of technologies to convert hypoxic microenvironments into hyperoxic microenvironments for tumour biology or therapy studies.

It is a major challenge to construct a long-lasting hyperoxic microenvironment in tumours due to the lack of constant and biocompatible oxygen sources. Considering that algal microbes are the major suppliers of $O_2$ on Earth, photosynthesis in algal chloroplasts could potentially be explored for $O_2$ supplements in tumours. The photosynthetic machinery requires a matched light source emitting 650–700 nm photons. Since rare earth-based upconversion nanoparticles (UCNPs) have shown an extraordinary capability to convert

[1]State Key Laboratory of Radiation Medicine and Protection, School for Radiological and Interdisciplinary Sciences (RAD-X), Collaborative Innovation Center of Radiation Medicine of Jiangsu Higher Education Institutions, Suzhou Medical College, Soochow University, Suzhou, Jiangsu 215123, China. [2]Department of Interventional Radiology, the First Affiliated Hospital of Soochow University, Soochow University, Suzhou, Jiangsu 215001, China. [3]Institute of Blood and Marrow Transplantation, National Clinical Research Center for Hematologic Diseases, Soochow University, Suzhou, China. [4]School of Public Health, Suzhou Medical College, Soochow University, Suzhou, Jiangsu 215123, China. [5]Division of NanoMedicine, Department of Medicine, California Nanosystems Institute, University of California, Los Angeles, CA 90095, USA. [6]These authors contributed equally: Weili Wang, Huizhen Zheng. ✉e-mail: liruibin@suda.edu.cn

biotransparent near-infra-red (NIR) lasers into visible light[10], these materials could be exploited to provide available photons in photosynthesis. We therefore hypothesized that a long-lasting hyperoxic microenvironment could be created by rational construction of algal microbes and UCNPs.

In this study, we pioneer a photosynthesis microcapsule (PMC) by encapsulating cyanobacteria and UCNPs in alginate microcapsules (MCs) that can be fabricated by an electrostatic droplet technique. Four cyanobacterial strains were subject to acclimation selection to acquire a suitable strain for the accommodation of physiological conditions. We comprehensively explore the impacts of NIR radiation, cell population and UCNP dose on $O_2$ production to design an optimized formula of PMCs. The impacts of hyperoxic microenvironments created by PMCs are examined in nine cancer cell lines and two tumour models, including orthotopic breast cancer in mice and transplanted hepatocarcinoma in rabbits.

## Results

### Bioengineering and characterization of the oxygenation capability of PMCs

To demonstrate our hypothesis, we first attempted to engineer the proposed PMC system, including the selection of algal microbes acclimatized to physiological conditions, synthesis of UCNPs compatible with photosynthesis, and encapsulation of algal microbes and UCNPs into MCs. Four parent algal microbes, *i.e.*, the spherical *Synechocystis sp.* 6803 (*S. sp.* 6803), *Chlorella ellipsoidea* (*C. ellipsoidea*), the rod-shaped *Synechococcus elongates* 7942 (*S. elongate.* 7942), and the boat-shaped *Scenedesmus obliquus* (*S. obliquus*), were selected in this research (Supplementary Fig. 1). Since these algal microbes grown in BG11 media at 25 °C cannot survive in mammalian cell culture media (e.g., DMEM) at 37 °C, they were subjected to acclimation procedure that allowed the cells to grow under physiological conditions by stepwise alternations of temperature (25 to 37 °C) and medium composition (BG11 medium to DMEM). Acclimatation could be assessed by assessment of cell density based on the absorbance of chlorophyll α at 650–700 nm[11]. As shown in Supplementary Fig. 2, while the proliferation of *S. obliquus* and *C. ellipsoidea* was significantly inhibited at temperatures >32 °C, *S. sp.* 6803 and *S. elongate.* 7942 were able to acclimate to the temperature increments. Stepwise changes from the BG11 medium to DMEM allowed us to acquire an evolved *S. sp.* 6803 (e-*S. sp.* 6803) strain, which maintained its activity at 37 °C in DMEM (Supplementary Fig. 3). This strain was therefore selected for PMC construction. We then synthesized a series of UCNPs with different

emissions by the crystal growth method[10]. An $Er^{3+}$ and $Yb^{3+}$-doped $NaYF_4$ nanorod (15.3 × 30.2 nm) was found to emit strong fluorescence at 660 nm, perfectly matching the absorbance of chlorophyll α (Supplementary Fig. 4 and Supplementary Fig. 5), which is the prominent component responsible for photosynthesis. Next, we engineered the PMCs by encapsulating algal microbes and UCNPs in the alginate-calcium microspheres under an electrostatic field via an electrostatic droplet generation system. As shown in Fig. 1, the whole process involves four steps: (i) homogenous mixing of algal microbes with UCNPs in alginate sodium; (ii) dispersing of alginate sodium solution into uniform droplets under an electrostatic field; (iii) encapsulating UCNPs and microbes by cross-linked alginate-calcium in $CaCl_2$ solutions; and (iv) coating of alginate-calcium microspheres by poly-L-lysine (PLL), a water-soluble polycation that is resistant to enzymatic degradation and capable of preventing microbe leakage (Supplementary Fig. 6). The resulting PMCs were examined by upconversion luminescence microscopy. While empty MCs and alga-encapsulated MCs had no fluorescence signals, PMCs emitted strong red fluorescence under 980 nm excitation (Fig. 2a). These results indicated that the encapsulated UCNPs thoroughly maintained their optical properties. We comprehensively examined the impacts of microbe density and UCNP concentration on the photosynthetic activity of PMCs (Fig. 2b). An optimized formula of PMCs (3 × 10³ algal cells and 0.67 μg of UCNPs per MC) was acquired for efficient oxygen production (1.6 μg/min). The oxygen generation of designated PMCs was dependent on the intensity and exposure time of NIR radiation, indicating a controllable oxygen supply (Supplementary Fig. 7). The encapsulated algal microbes in PMCs survived in DMEM for over 1 month (Supplementary Fig. 8).

Then, we examined the oxygenation capability of PMCs in physiological solutions as well as tumours. PMCs were first incubated in DMEM at 37 °C. Oxygen is often generated along with other oxygen species in the photosynthesis of chloroplasts[12]. The production of oxidative radicals in PMCs was examined by 2′,7′-dichlorodihydrofluorescein (DCFH) staining[13] to assess photosynthetic activity. Nonfluorescent DCFH could be oxidized into 2′,7′-dichlorofluorescein (DCF), emitting green fluorescence under hyperoxic conditions. As shown in Fig. 2c, intense fluorescence signals were detected in fully constructed PMCs, whereas alga-free and UCNP-free PMCs showed limited signals. The released oxygen in the medium was continuously monitored for 24 h. As shown in Supplementary Fig. 9, the PMCs could stably synthesize oxygen, which reached 6.2 ± 0.5 mg/L at 1 h post-NIR exposure and slowly declined to 3.0 ± 0.3 mg/mL in 24 h. Given the

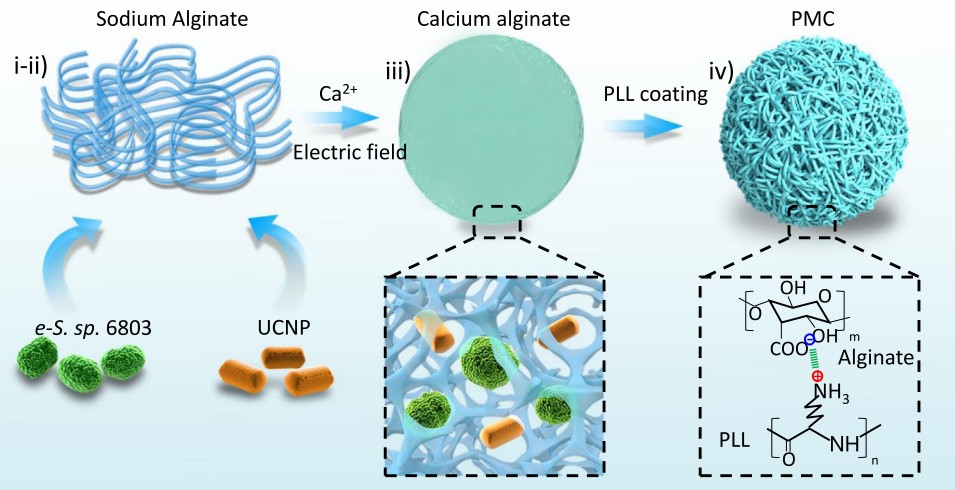

**Fig. 1 | Schematic displaying the critical steps of PMC construction.** (i) homogenous mixing of algal microbes with UCNPs in alginate sodium; (ii) dispersing of alginate sodium solution into uniform droplets under an electrostatic field; (iii) encapsulating UCNPs and microbes by cross-linked alginate-calcium in $CaCl_2$ solutions; and (iv) coating of alginate-calcium microspheres by poly-L-lysine (PLL).

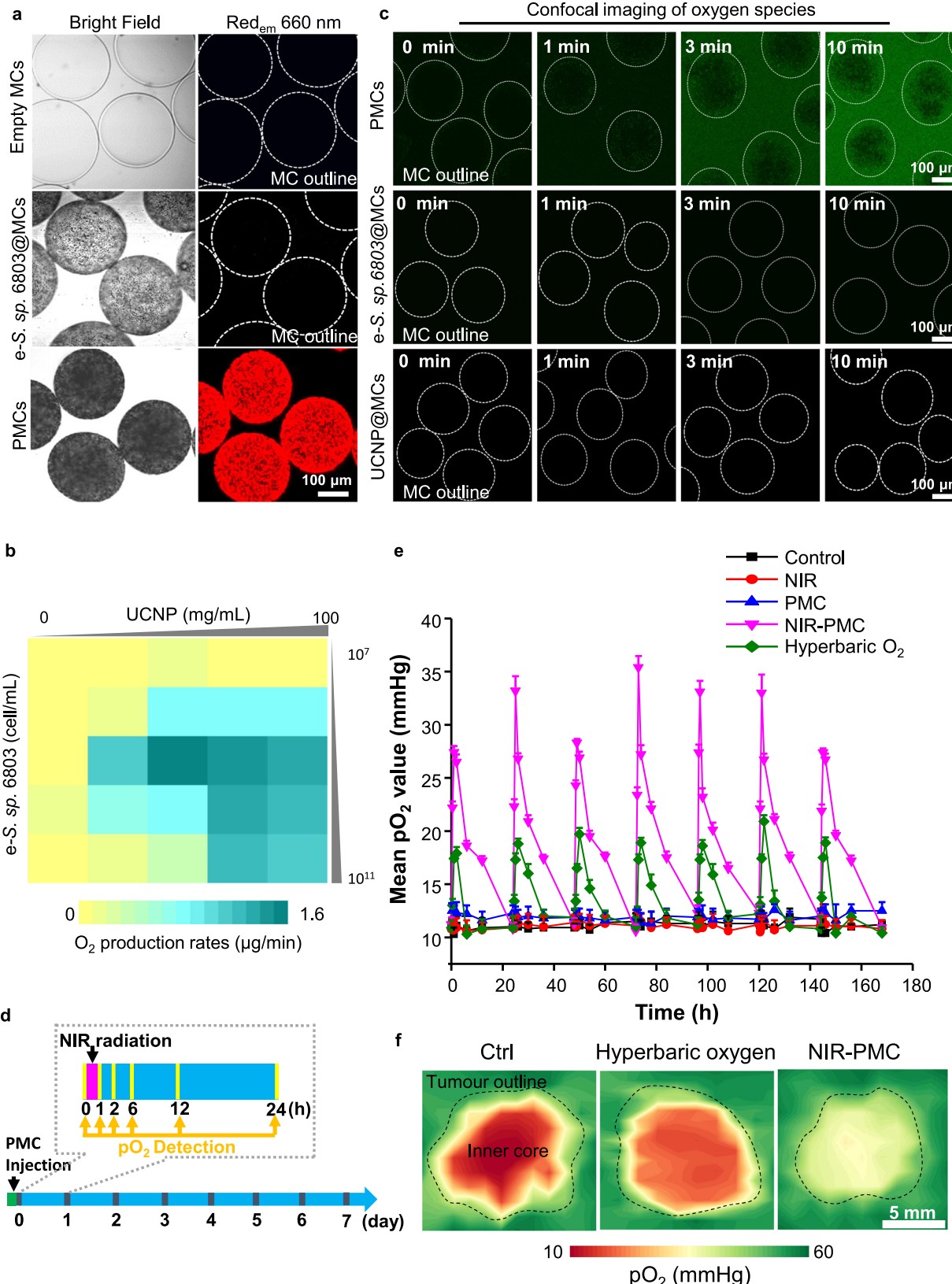

interstitial pressures at 20–40 mmHg in solid tumours[14–16], their impacts on photosynthesis of PMCs were examined. As shown in Supplementary Fig. 10, we designed a device consisting of a sealed 10 mL syringe, calibrated weights and Clark electrodes for dynamic detection of partial pressure of oxygen ($pO_2$) at different pressures. While the $pO_2$ in DMEM reached $105.7 \pm 6.1$ mmHg at normal

atmospheric pressure, the extra pressure had little effect on oxygenation profiles of PMCs. In addition, the oxygenation of PMCs was assessed in breast tumour. As shown in Fig. 2d, e, $pO_2$ in the core (~5 mm to the outer layer) of tumours reached peaks ($27.2–35.4$ mmHg) at 1 h post-NIR exposure and slowly declined to basal levels of $10.8–12.3$ mmHg at 24 h. The oxygenation curves displayed similar

**Fig. 2 | Bioengineering and characterization of PMCs. a** Imaging of constructed PMCs. The empty MCs, MCs containing algal cells (e-*S*. sp. 6803@MCs) and fully constructed PMCs were subjected to imaging by upconversion luminescence microscopy at 980 nm. The edge of MCs is described by white outline. **b** A heatmap presenting O$_2$ production by PMCs with different formulations. Algal cells at 10$^7$–10$^{11}$ cell/mL, UCNPs at 0–100 mg/mL, and 1 mL of alginate sodium were mixed together to prepare different formulations of PMCs by an electrostatic droplet generation system. The resulting PMCs were exposed to 900 mW/cm$^2$ NIR radiation for 20 min to measure O$_2$ levels by a portable dissolved oxygen metre. **c** Visualization of synthesized oxygen species in PMCs by DCFH staining. PMCs, e-*S*. sp. 6803@MCs and UCNP@MCs cultured in the dark for 12 h were incubated with 10 μg/mL DCFH for 10 min and then exposed to 300 mW/cm$^2$ 980 nm NIR radiation

for 0, 1, 3 and 10 min. The treated microspheres were immediately visualized by confocal microscopy at 488 nm excitation. The edge of MCs is described by white outline. **d** Schematic image to show pO$_2$ detection in tumours. **e** Measurements of the partial pressures of oxygen (pO$_2$) in the cores of tumours at different time points. Data are presented as means ± SD derived from three tumours. **f** visual display of pO$_2$ in tumours. A Clark oxygen electrode was used to measure pO$_2$ in mouse breast tumours received saline, PMC, NIR, hyperbaric oxygen (60%) and NIR-PMC treatments. The pO$_2$ values were recorded in tumours at 5 mm depths for 7 days or different depths of tumours at 1 h post NIR-PMC treatments. The pO$_2$ values at the largest cross-section of tumours were integrated by Python for constructions of the heatmaps. The edge of tumour is described by black outline. Source data are provided as a Source data file.

patterns in seven cycles of NIR exposure, indicating a stable photosynthetic activity of PMCs in tumours. Compared to hyperbaric oxygen, the pO$_2$ in tumour cores reached 6.7-fold higher levels in NIR-PMC treatment. Heatmaps were constructed to visually assess the pO$_2$ from the tumour outer layer to the inner core (Fig. 2f). While untreated tumours (control) had severe hypoxia in deep tumours, yellowish-green colours in NIR-PMC-treated tumours suggested significant elevations in oxygen levels. These results thoroughly documented the oxygenation capability of PMCs in vitro and in vivo.

## Proliferation of tumour cells under hyperoxic conditions by PMCs

We then wondered whether O$_2$ supply by PMCs may impact the proliferation of cancer cells. The effects of PMCs were assessed in nine cancer cell lines, including breast (MCF-7, 4T1), lung (A549), liver (HepG2, VX2), intestinal (HCT116), pancreatic (PANC-1), stomach (MGC-803) and cervical (HeLa) cells, across three mammalian species (human, mouse and rabbit) and three primary cell lines (mouse embryonic fibroblasts (MEFs), heart and kidney cells). The cell metabolic activities were examined by a colorimetric substrate in an MTS assay. To prevent the cytotoxic effects of heat in NIR radiation, the NIR exposure dose was set at 300 mW/cm$^2$ for 20 min (Supplementary Fig. 11). Three intervals of NIR radiation were applied to PMCs in 24 h culture to create a hyperoxic biocontext with oxygen levels of 1.8–5.3 mg/L. As shown in Fig. 3a and Supplementary Fig. 12, PMCs coupled with NIR (NIR-PMCs) had little cytotoxicity in three normal cell lines but inhibited the proliferation of eight cancer cell lines. Notably, the oxygen-sufficient media completely suppressed the duplications of HCT116, HepG2, MCF-7 and 4T1 cells. We comprehensively examined the impacts of PMC integrates in detail, including MCs, algae and UCNPs, on MCF-7 and HepG2 cells with or without NIR radiation. As a result, the suppression effect was mainly detected in cells under NIR-PMC treatment (Fig. 3b and Supplementary Fig. 13). To further demonstrate this point, we visualized the new daughter cancer cells by a BeyoClick-iT® EdU-594 proliferation assay kit consisting of EdU (5-ethynyl-2′-deoxyuridine) that could be incorporated into newly synthesized DNA and fluorescently labelled by a specific click reaction[17]. As shown in Fig. 3c, NIR-PMC treatment significantly suppressed the proportion of daughter cells divided from breast cancer (MCF-7) and hepatocarcinoma (HepG2) cells, as evidenced by the reduction in EdU-positive cells (red). Notably, the hyperoxia media became invalidated once the dissolved oxygen was removed by N$_2$ purging (Supplementary Fig. 14), suggesting that oxygen rather than metabolites from algal cells were responsible for the suppression effect of NIR-PMCs. Flow cytometry analysis indicated few apoptotic cell deaths (Supplementary Fig. 15).

Since NIR-PMCs had a limited effect on primary cells but significantly suppressed the proliferation of cancer cells, we speculated that hyperoxia may specifically impact the signals in carcinogenesis pathways. Given that nuclear factor kappa B (NF-κB), phosphoinositol 3′-kinase (PI3K), protein kinase B (AKT), mammalian target of rapamycin (mTOR), mitogen-activated protein kinase (MAPK) and extracellular

signal regulated kinase (ERK) play crucial roles in cancer cell survival and proliferation[18–21], they were examined in HepG2 cells by immunoblotting. Endothelial growth factor (EGF), insulin and TNF-α were exploited as inducers[22–24], and 3-methyladenine (3-MA), MK-2206, rapamycin, PD98059 and BMS-345541 (BMS) were included as inhibitors[25–30]. As shown in Fig. 3d and Supplementary Fig. 16, significant suppression of phosphorylated IκBα (p-IκBα) was detected in PMC-treated cells. In contrast, NIR-PMCs had little impact on the expressions of ERK, PI3K, AKT and mTOR. These results indicated that the effect of NIR-PMCs on inhibiting proliferation was mainly associated with the NF-κB pathway. To further confirm the role of NF-κB, we attempted to alter NF-κB activity in the presence or absence of PMCs to examine cell proliferation by staining of new daughter cells (Supplementary Fig. 17). The proliferation of cancer cells could be inhibited by BMS but enhanced by TNF-α in the absence of PMCs. Notably, in the presence of PMCs, while BMS displayed a stronger inhibitory effect on cell growth, TNF-α failed to boost cell duplications.

Substantial studies have shown that hypoxia is responsible for the immune suppression of the tumour microenvironment by hypoxia-induced HIF-1α and adenosine release in cancer cells and the inhibition of immune cell function by increasing the amount of adenosine A2A receptors (A2ARs)[31–33]. We speculated that hyperoxia by NIR-PMCs may boost immune activity by eliminating the hypoxia/HIF-1α/adenosine/A2AR axis. To test this speculation, we examined the effects of NIR-PMCs in cancer cell spheroids constructed by encapsulation of HepG2 or MCF-7 cells in Matrigel. First, the hypoxyprobe FITC-MAb1 was used to assess hypoxia status in cancer cell spheroids. As shown in Fig. 4a, intense green fluorescence was observed in untreated spheroids, indicating severe hypoxia, whereas NIR-PMC treatment elevated oxygen levels and ameliorated the hypoxia status. Quantification of hypoxia-inducible factor-1α (HIF-1α) in cell lysates by ELISA indicated that NIR-PMC treatment dramatically reduced the levels of HIF-1α from 335.7 to 148.3 pg/mg protein in HepG-2 cell spheroids and 614.6 to 107.9 pg/mg protein in MCF-7 cell spheroids (Fig. 4b). Consequently, adenosine productions had 42.9% and 71.7% declines in MCF-7 and HepG-2 cell spheroids, respectively (Fig. 4c). IFN-γ production[34,35] was examined to assess the effector function of T cells after hyperoxia treatment. Whereby phorbol myristate acetate (PMA) was included as the inducers of T cell activations. As shown in Fig. 4d, e, in the absence of PMA, T cells pretreated with or without NIR-PMCs only produced negligible levels of IFN-γ. In contrast, the percentage of IFN-γ-producing T cells in NIR-PMC treatment group was remarkably elevated to 13.5% in the presence of PMA, which was about 2.6-fold higher than that without NIR-PMC pretreatment. For testing the potential effect of NIR-PMCs on the cytolytic activity of NK cells, MHC-devoid mouse lymphoma cell line Yac-1 highly sensitive to NK cells was used as the target cells[36,37]. As shown in Fig. 4f, after co-culture for 6 h, up to 95.6% of Yac-1 cells were eliminated by NK cells in NIR-PMC treatment, and in stark contrast, only of 58.3% Yac-1 cells were killed in the control groups. These results indicated the reinvigoration of effector function of T and NK cells in hyperoxic environment created by NIR-PMCs.

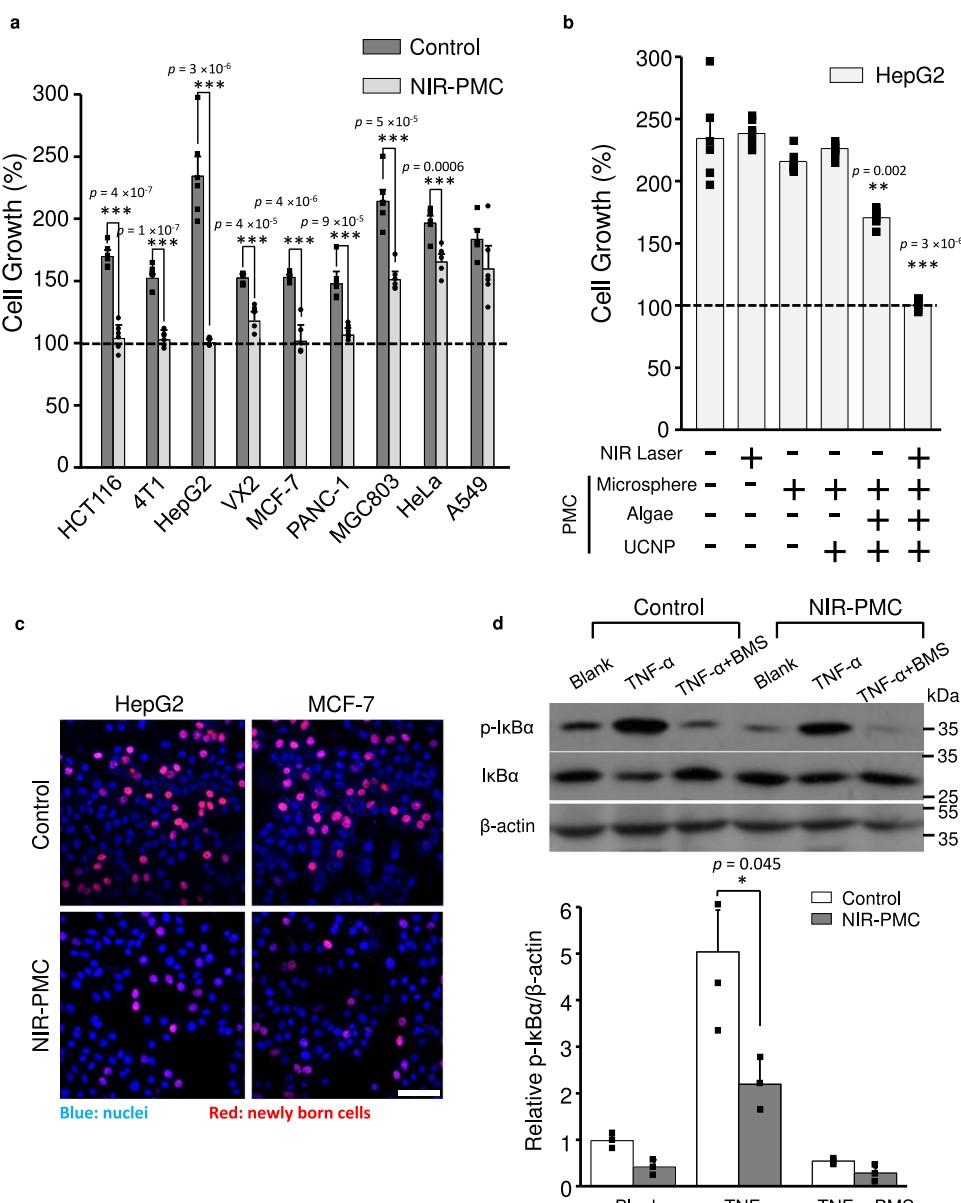

**Fig. 3 | Impacts of NIR-PMCs on cancer cell proliferation. a** Cell proliferation assessment. HCT116, 4T1, HepG2, VX2, MCF-7, PANC-1, MGC-803, HeLa and A549 cells incubated with or without PMCs (25 μL, 3.6 × 10⁴/mL) were exposed to 300 mW/cm² NIR radiation for three intervals. After 24 h, cell viability was examined by MTS assays ($n = 6$). The data are presented as mean ± SD. ***$p < 0.001$ compared to control cells according to two-tailed Student's $t$-test. **b** Impacts of integrates in PMCs. HepG2 cells were incubated together with MCs, UCNP@MCs, e-*synechocystis*@MCs and PMCs in the presence or absence of NIR radiation for cell viability assessment ($n = 6$). The data are presented as the mean ± SD. ***$p < 0.001$ and **$p < 0.01$ compared to the control group according to two-tailed Student's $t$-test. **c** Confocal visualization of new daughter cells. HepG2 and MCF-7 cells after

NIR-PMC treatment were stained with a BeyoClick-iT® EdU-594 assay kit (scale bar: 20 μm) for confocal imaging. **d** The cellular effects of NIR-PMCs on the NF-κB signalling pathway. HepG2 cells were pretreated with NIR-PMCs (300 mW/cm² NIR radiation for three intervals). After 24 h, all the cells were pretreated with 20 nM BMS and 2 μL of dimethyl sulfoxide (DMSO) (blank) for 2 h prior to stimulation with 200 ng/mL TNFα for 10 min. Cell lysates were collected and subjected to Western blotting analysis of IκBα and phosphorylated IκBα ($n = 3$, the samples derive from the same experiment and that blots were processed in parallel). The ratios of p-IκBα $v.s.$ β-actin were determined by ImageJ 1.51. The data are presented as the mean ± SD. *$p < 0.05$ compared to control cells according to two-tailed Student's $t$-test. Source data are provided as a Source data file.

## Effects of oxygen supply by PMC implants on hepatocarcinoma

Next, we examined the in vivo impacts of oxygen supplements on hepatocarcinoma by in situ implantation of PMCs in the livers of rabbits. To determine a safe and effective radiation dose in vivo, we examined the impacts of NIR radiation on skin pathological changes as well as oxygen generation. An NIR radiation dose of 900 mW/cm² for 20 min was selected for animal treatment because it created a hyperoxic microenvironment (2.7–3.4 mg/L) and prevented hyperthermia-induced tissue damage (Supplementary Fig. 18). More than 65% of cyanobacteria in PMCs survived in the liver for at least 1 months

(Supplementary Fig. 19). As indicated in the treatment scheme (Fig. 5a), hepatocarcinoma-bearing rabbits that received an intratumour injection of PMCs (500 μL, 3.6 × 10⁴/mL) were exposed to 980 nm NIR radiation for 42 days. NIR-PMC treatment created a long-lasting hyperoxic tumour microenvironment, evidenced by a dramatic reduction in HIF-1α expression (Fig. 5b). Consequently, the downstream targets of HIF-1α, including BCL2/adenovirus E1B protein-interacting protein 3 (BNIP-3), vascular endothelial growth factor (VEGF), insulin-like growth factor (IGF-2), matrix metalloproteinase-2 (MMP2) and transforming growth factor-alpha (TGF-α), significantly

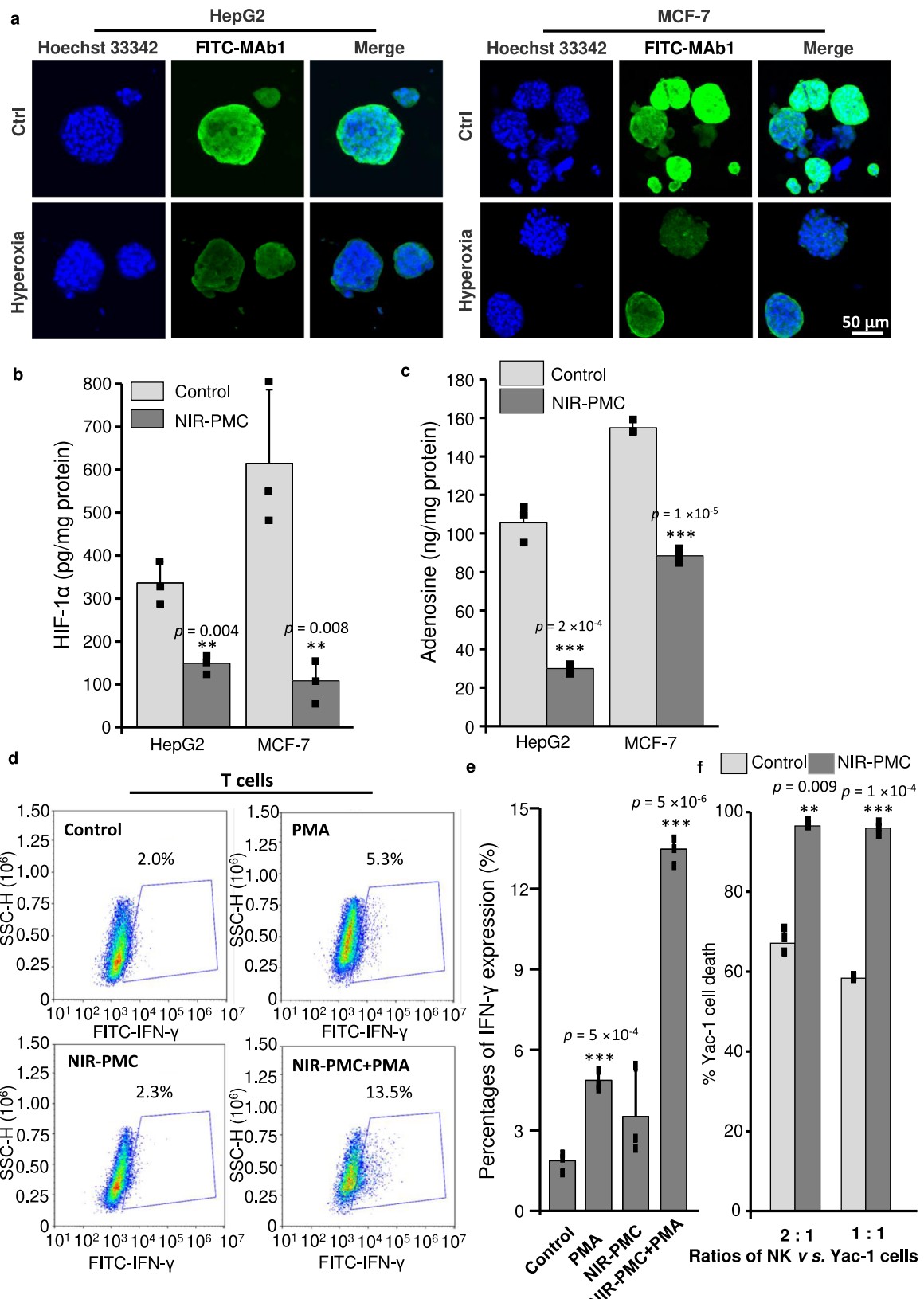

decreased after NIR-PMC treatment (Supplementary Fig. 20 and Supplementary Table 2). These targets play crucial roles in cancer cell survival, proliferation and metastasis, and their declines imply suppression of tumour growth[38–41].

Tumour growth was examined via CT imaging for 42 days (Fig. 5c). While the tumour sizes in untreated rabbits rapidly increased to 4 cm³

on the 14th day and 16 cm³ on the 28th day, NIR-PMC treatment completely inhibited tumour growth in 14 days and displayed a slow increase in tumour sizes to 4 cm³ on the 28th day (Fig. 5d). Since the lung is the most common target organ for hepatocarcinoma metastasis[42], we also examined this organ by CT imaging. As shown in Fig. 5e, notable signs of metastases could be differentiated in the lungs

**Fig. 4 | Assessment of the hypoxia/HIF-1α/adenosine/A2AR axis in tumour organoids. a** Confocal imaging of hypoxic status. HepG2 and MCF-7 cells were incorporated in matrigel to prepare spheroids. After 7 days incubation, the constructed spheroids were exposed to 3600/mL PMCs and received 300 mW/cm² NIR radiation for three intervals. The treated spheroids were subjected to staining by HypoxyprobeTM−1 Plus Kit (green) and Hoechst 33342 (blue). **b** HIF-1α and **c** adenosine productions in cancer cell spheroids (*n* = 3). The cells in spheroids were collected and lysed for HIF-1α or adenosine detection by ELISA and LC-MS, respectively. **d** Representative images and **e** quantification of IFN-γ expression in T cells by flow cytometry analysis (*n* = 3). T cells were treated with 50 ng/ml PMA,

and 1 μL/mL Golgi inhibitor for 18 h with/without NIR-PMC for three intervals. The treated cells were fixed, permeabilized and stained by anti-IFN-γ-FITC for flow cytometry analysis. **f** Impacts of hyperoxia on the cytolytic activity of NK cells (*n* = 3). The sorted NK cells (2 × 10⁵ cells/well) were mixed with Yac-1 cells at the ratios of 1:1 or 2:1 in 24-well plates, and co-cultured with/without PMCs, followed by three intervals of NIR exposure. After 6 h incubation, the cell mixtures were collected to lyse Yac-1 cells. The luminescence of cell lysates was measured by a Microplate reader. Data are expressed as mean ± SD. **$p < 0.01$ and ***$p < 0.001$ compared to Ctrl cells by two-tailed Student *t*-test. Source data are provided as a Source data file.

---

of untreated tumour-bearing rabbits at 14 d, and a large number of tumour foci (51 ± 25/lung) formed on the 28th day. Notably, there were few detectable pulmonary metastases in NIR-PMC treatments even on the 42nd day (Fig. 5f, Supplementary Fig. 21). Consistently, liver sections of rabbits receiving NIR-PMC treatment displayed a dramatic reduction in the tumour metastasis biomarker vascular cell adhesion molecule-1 (VCAM-1)[43] (Supplementary Fig. 22). Kaplan–Meier plots for survival rate comparisons were generated by continued daily monitoring of the animals at the point of moribund status or spontaneous death. Log rank testing demonstrated a statistically significant survival benefit ($p < 0.05$) for the NIR-PMC treatment compared to the vehicle control treatment (Fig. 5g). Notably, two out of ten rabbits lived >140 days and were almost cured, as there were no detectable tumour nodules in the livers and lungs by CT imaging and ex vivo examination (Supplementary Fig. 23). To our surprise, luminescent PMCs could still be observed in the liver and maintained spherical morphologies (Supplementary Fig. 23). These results indicated that the hyperoxic microenvironment created by NIR-PMCs could greatly slow tumour progression, inhibit tumour metastasis and enhance the survival rates of hepatocarcinoma-bearing rabbits.

## Combined treatment of breast cancer by PMC implants and immunotherapy in mice

Since oxygen species have been widely reported as prerequisites of immune activation[44], we speculated that long-lasting oxygen supplements by NIR-PMCs may facilitate immunotherapy by activating adaptive immune responses in the tumour microenvironment. We verified this by LC-MS analysis of adenosine and immunostaining of CD4, CD39, CD206 and CD73 expression in an orthotopic breast tumour model by inoculating firefly luciferase-transfected 4T1 (fLuc-4T1) tumour cells into the breast pads of BALB/c mice. As shown in Fig. 6a, b, NIR-PMC treatment significantly ameliorated HIF-1α and adenosine production, enhanced CD4 expression, and reduced CD39, CD206, CD73 and A2AR expressions in tumour sections; this indicated that T helper cells, B cells, and NK cells were activated and that differentiation of M1 macrophages in the tumour microenvironment occurred[45,46]. Therefore, a synergistic therapeutic effect could be expected for a combined treatment of NIR-PMCs and checkpoint inhibitors. The tumour-bearing mice screened via bioluminescence imaging (BLI) were randomly separated into four groups: no treatment (vehicle control), NIR-PMC treatment, anti-PD-1 treatment (200 μg/mouse twice a week) and cotreatment with NIR-PMCs and anti-PD-1. Figure 6c displayed the scheme of combined treatment. As shown in Fig. 6d, rapid tumour growth was detected in untreated tumour-bearing mice by the intensive luminescence of 4T1 cells as well as in the bright field images. The fast proliferation of breast cancer cells led to nutrient deficiency and cell necrosis, evidenced by the diminishing luminescence in tumour cores. While cotreatment with NIR-PMCs and anti-PD-1 displayed a potent therapeutic effect, NIR-PMC and anti-PD-1 treatments merely suppressed tumour growth in the first 2 weeks but subsequently grew rapidly. This result was further confirmed by the tumour volume data. The tumour sizes in cotreatment were three-fold smaller than those in anti-PD-1 treatment (Fig. 6e, Supplementary Fig. 24). Tumour metastases were also examined in animal lungs. As shown in Fig. 6f, while anti-PD-1 had a

limited effect on tumour metastases, NIR-PMC treatment significantly reduced tumour nodules in dissected lungs. Notably, there were no detectable pulmonary tumour metastases in the cotreatment group. We also compared the survival rates of animals in the different treatments (Fig. 6g). All untreated tumour-bearing animals were either dead or reached moribund status in 25 days, whereas NIR-PMC or anti-PD-1 treatments significantly prolonged animal survival (+40%). Interestingly, no animal deaths were observed after cotreatment for 50 days, and 8 out of 10 mice survived for more than 60 days. Overall, these data suggested that the combination of PMC implants with immune checkpoint inhibitors showed a strong synergistic effect (CI = 0.23) in mice bearing breast cancer.

To assess the biosafety of NIR-PMCs in animals, PMCs were administered to healthy mice by intraperitoneal or subcutaneous injection and exposed to NIR radiation. The animal tissues were collected for histological examinations. As shown in Supplementary Fig. 25, H&E staining showed negligible pathological changes in all tested organs, and TUNEL staining indicated no necrotic cell death in skin sections. Although blood test results showed limited changes of blood platelets, neutrophils and monocytes at the 1st or 3rd day post injections, they all recovered at the 8th day. These results well demonstrated the biosafety of PMCs (Supplementary Table 4 and Supplementary Table 5).

## Discussion

Tumour therapeutic strategies rely on our understanding of carcinogenesis mechanisms. While somatic selection is often recognized as the prominent cause of carcinogenesis, substantial amounts of research have aimed at the discovery of chemotherapeutics or physicotherapeutics to eliminate mutant cells[7,47]. Although millions of cancer patients have successfully extended their lives because of the killing of tumours, side effects, drug resistance[48] and tumour metastasis[49] have become formidable hindrances. This has necessitated a rethinking of cancer therapeutic strategies. Substantial research interests have been aroused to improve hypoxic tumour microenvironments. A hyperbaric chamber is a conventional facility in clinics to elevate blood oxygen levels. It was discovered by Joseph Priestley in 1775, but the first therapeutic application was conducted in decompression sickness 150 years later[50]. Afterwards, these facilities were also used for cardiac surgery[51], infectious diseases[52] and tumour therapies[53]. However, the low specificity to target tissue and poor efficiency of hyperbaric oxygen limit its applications in cancer patients[53]. To overcome these limitations, drug carriers were explored to specifically deliver oxygen into tumours. For instance, Liang et al. designed O₂@PFOB@PGL nanoparticles, which could act as prominent oxygen reservoirs and effectively deliver oxygen to hypoxic tumours to enhance photodynamic therapy[54]. Platinum-based nanoparticles could react with endogenous H₂O₂ to generate O₂ inside the tumour site and improve the hypoxic microenvironment for combination therapy[55,56]. Perfluorocarbons (PFCs), synthetic materials displaying a high loading capacity of oxygen, were able to unload oxygen in a hypoxic tumour microenvironment[57]. However, these approaches could merely provide transient O₂ supplements. Herein, we constructed a smart micro-oxygen factory by encapsulation of UCNPs

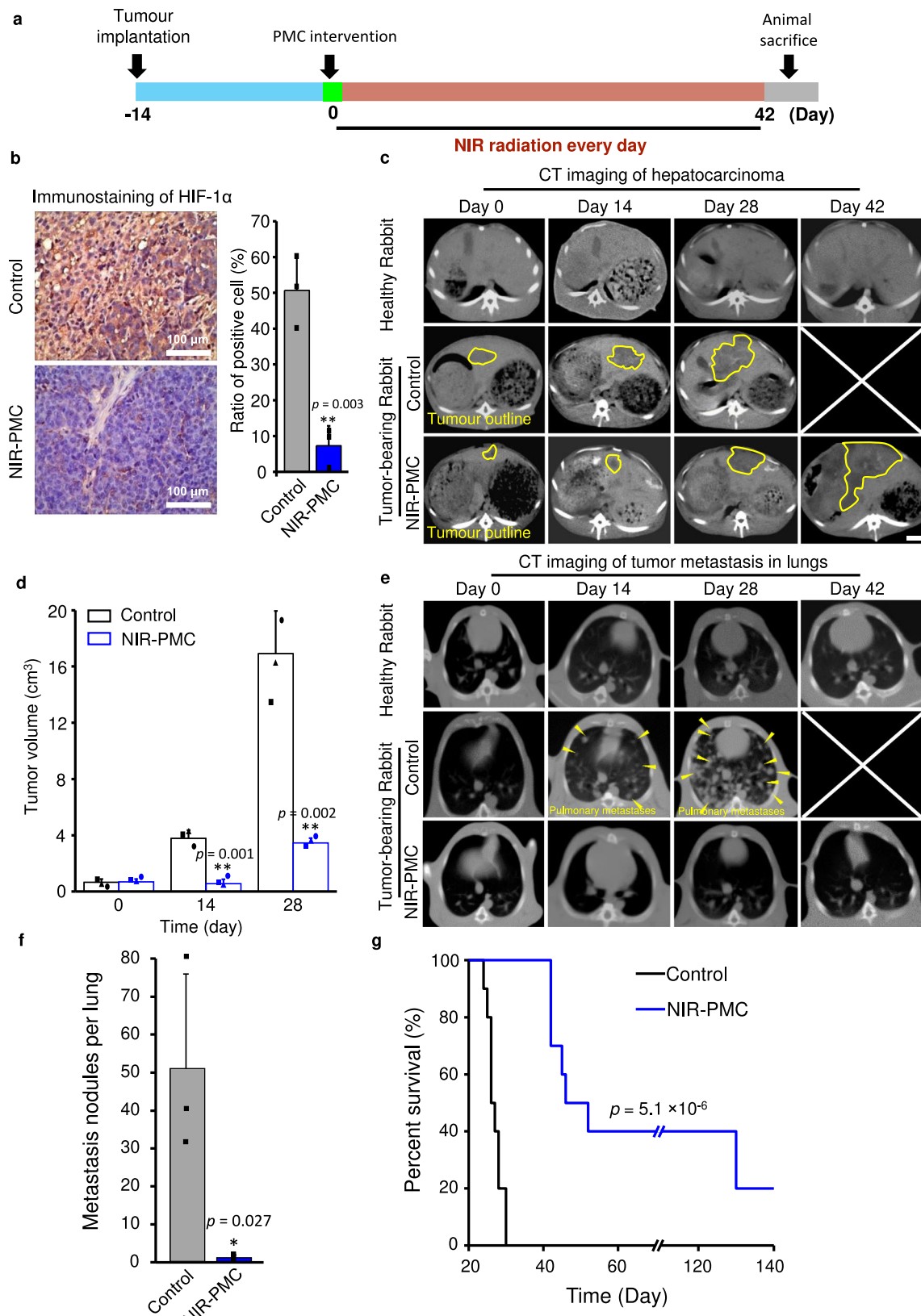

and algae in microcapsules. Unlike the reported oxygen delivery strategy, the resulting PMCs exploited NIR lasers to control the photosynthesis of algae and allow long-lasting oxygen supplements. Although the NIRs are applied to animals daily, the $pO_2$ in tumours reached peaks (27.2–35.4 mmHg) at 1 h post-NIR exposure and returned back to hypoxic status (10 mmHg) in 24 h. The periodic

oxygen supplementation may impact the metabolic network and phenotypes of cancer cells, which should be considered in follow-on studies. In this regard, long-lasting antagonism of hypoxic microenvironments in tumours may suppress cell duplication and slow tumour progression. Our findings in hepatocarcinoma tumour-bearing rabbits supported this hypothesis, as no tumour-killing

**Fig. 5 | Effects of the hyperoxic microenvironment by PMC implants in rabbit hepatocarcinoma. a** Illustration of NIR-PMC treatment in hepatocarcinoma tumour-bearing rabbits. At 14 days post-tumour implantation (tumour sizes are ~1 cm³), 500 μL of PMCs ($3.6 \times 10^4$/mL) were intratumourally injected into rabbits, followed by 900 mW/cm² NIR radiation for 42 days. **b** Immunostaining of HIF-1α in hepatic tumours. After 28 days, the rabbits were sacrificed to collect tumour tissues. The tissue samples were fixed in 4% formalin for immunohistochemical staining. The ratios of HIF-1α-positive cells (left panel) were calculated by a digital pathology scanner (DMetrix). The data are presented as the mean ± SD derived from three images of independent rabbit liver tumours. **$p < 0.01$ compared to vehicle control according to two-tailed Student's $t$-test. **c** Representative axial CT images of tumour growth. Tumour-bearing rabbits were treated with or without NIR-PMCs. Healthy, tumour-bearing and NIR-PMC treated animals were subjected to CT imaging at 0, 14, 28, and 42 days. The shape/edge of hepatocarcinoma tumours are described by yellow outline (Scale bars, 1 cm). **d** Measurement of tumour sizes by CT images. The sizes of hepatocarcinoma tumours were measured by Neusoft PACS/Ris V5.5 at 28 days post-treatment ($n = 3$). The data are presented as the mean ± SD. **$p < 0.01$ compared to the data for untreated hepatocarcinoma tumour-bearing rabbits according to two-tailed Student's $t$-test. **e** Representative axial CT images of pulmonary tumour metastases. The animal lungs were imaged by CT at 0, 14, 28, and 42 days post-treatment. The nodules of pulmonary metastases are described by yellow arrows. **f** The numbers of pulmonary metastasis nodules in rabbit lungs ($n = 3$). The data are presented as the mean ± SD. *$p < 0.05$ compared to untreated animals according to two-tailed Student's $t$-test. **g** Kaplan–Meier curves showing the survival rates of the indicated rabbits at 140 days ($n = 10$). The data are presented as *$p < 0.05$, compared to the data for untreated hepatocarcinoma tumour-bearing rabbits according to Log rank test by SPSS statistics 17.0 software. Source data are provided as a Source data file.

agents were included. Notably, in all ten tested animals, two rabbits were cancer free, as they lived five times longer (>140 days) than the untreated animals did (average survival time ~27 days) and had no detectable tumour nodes. However, these findings may need more validation across different tumour models.

Although surgery is the first choice for 30–40% of cancer patients at every early (Stage 0) or early (Stage I) stage[58,59], PMCs may be administered alone or may act synergistically with the effects of transarterial chemoembolization (TACE) and interventional, immune and targeted therapies in cancer patients at intermediate or late stages (>Stage II) to establish local control and palliation. The PMCs were deliberately designed to accommodate the interventional device. Although intratumor injection was selected for the administration of PMCs in animals, PMCs could be applied in human patients by transcatheter arterial chemoembolization. To facilitate future clinical applications, the dose and duration time of NIR radiation should be carefully examined. Taking hepatocarcinoma as an example, 900 mW/cm² NIR radiation at 980 nm was applied to rabbits for 60 min/day. Given that the depth of hepatocarcinoma in rabbits is 0.3–0.7 cm, the tumour received 300–500 mW/cm² radiation after penetration of the animal belly[60]. To acquire such excitation intensity in human patients, the radiation dose has to be increased to 2000 mW/cm² or the duration time should be extended to 180 min because hepatocarcinoma in human patients are often deeper than they are in rabbits[61–63] and because the intensity of NIR radiation at 980 nm would decline to <30% after penetration of 0.7–1.0 cm belly tissues in humans. However, this amount of NIR radiation may induce hyperthermia damage. Alternatively, NIR-II radiation at 1200–1700 nm would be more suitable for clinical applications, as NIR-II photons have a much deeper penetration capability than NIR lasers at 980 nm[64]. UCNPs with excitations in the NIR-II region could be exploited to construct PMCs for potential applications in clinics. Since interventional therapy is a conventional treatment in hepatocarcinoma patients, PMCs are a promising implant for clinical applications.

In addition to the hepatocarcinoma model, we examined the effects of PMCs in breast cancer-bearing mice. The effects of different constituents in PMCs were comprehensively examined. Consistent with the in vitro results (Fig. 3b), therapeutic effects were mainly observed in animals that received NIR-PMC treatment (Supplementary Fig. 26 and Supplementary Fig. 27). The bio-distribution of PMCs in tumours were examined by visual observation of the luminescence, and their locations had little changes (Supplementary Fig. 28). Oxygen has been reported to activate antitumour immune responses, such as the differentiation of M2 tumour-associated macrophages (TAMs) to M1[65] and the activation of T cells[66] and NK cells by the synthesis of superoxide radicals[67]. Hatfield et al. proposed eliminating the hypoxia/HIF-1α/adenosine/A2AR axis by oxygenating agents to improve cancer immunotherapy or synergize other oncological treatments[31,44]. Our study demonstrated that NIR-PMCs could significantly improve the hypoxia status and inhibit HIF-1α and

adenosine production in cancer cell spheroids and breast tumour. In addition, hyperoxia by NIR-PMCs could reinvigorate the effector function of T and NK cells in terms of IFN-γ production and cytolytic capacity in vitro. NIR-PMC-treated tumours showed lower A2AR expression but enhanced CD4 expression, suggesting the occurrence of enhanced anti-cancer immune response in breast tumours. The combination of NIR-PMCs with immune checkpoint inhibitors showed considerable survival benefits in breast tumour-bearing mice. These results strongly supported the proposed therapeutic strategy by Hatfield et al.

Tumour metastasis is a complex process involving the dissemination of cancer; epithelial–mesenchymal transition; and invasion by collective migration, cancer cell circulation, interaction with immune cells, metastatic colonization, etc. A few biomolecules, including FAT1[68], ribosomal proteins[69], the transmembrane protein CCDC25[70], Nrf2[71] and E-cadherin[72], were recently found to regulate metastasis progression. In addition, as a potent microenvironmental factor, hypoxia was reported to promote multiple steps (e.g., invasion, migration, intravasation and extravasation) within the metastatic cascade[1]. Although tumour metastasis is the major cause of treatment failure in cancer patients and is responsible for 90% of cancer-related mortality[73], there are few clinical interventions. However, the in vivo oxygen factory constructed in this study could be potentially exploited in cancer treatments for suppression of tumour metastasis. PMC implants dramatically prevented tumour metastasis in orthotopic breast cancer and transplanted hepatocarcinoma models. The hyperoxic microenvironments created by NIR PMCs may block cancer cell invasion by antagonizing metastasis signals[74].

In summary, we successfully constructed oxygen factories (PMCs) by encapsulating cyanobacteria and UCNPs in alginate-polylysine microcapsules. PMCs enabled photosynthesis under NIR radiation and provided long-lasting and controllable oxygenation in biological contexts. The supplemented O₂ by PMCs inhibited HIF-1α production and NF-κB activation in cancer cells and dramatically prevented cancer cell duplication in a nontoxic manner. In vivo hyperoxic microenvironments were created by implants of PMCs in orthotopic hepatocarcinoma and breast tumours and significantly inhibited tumour growth and metastasis. A potent synergistic effect was demonstrated in breast cancer of mice receiving cotreatment with PMC implants and a checkpoint inhibitor (anti-PD-1). Overall, our findings demonstrated the suppressive effect of the hyperoxic microenvironment on tumour progression.

## Methods
### Reagents and materials
Antibodies for Western-blot in this study including anti-phospho-IκBα (Cat. 9246), anti-IκBα (Cat. 4814), were purchased from Cell Signalling Technology Inc. (Boston, MA, USA). 3-Methyladenine (HY-19312), MK-2206 (HY-10358), Rapamycin (HY-10219) and PD98059 (HY-12028) and IL-2 (HY-P7077) were purchased from MedChemExpress (New Jersey,

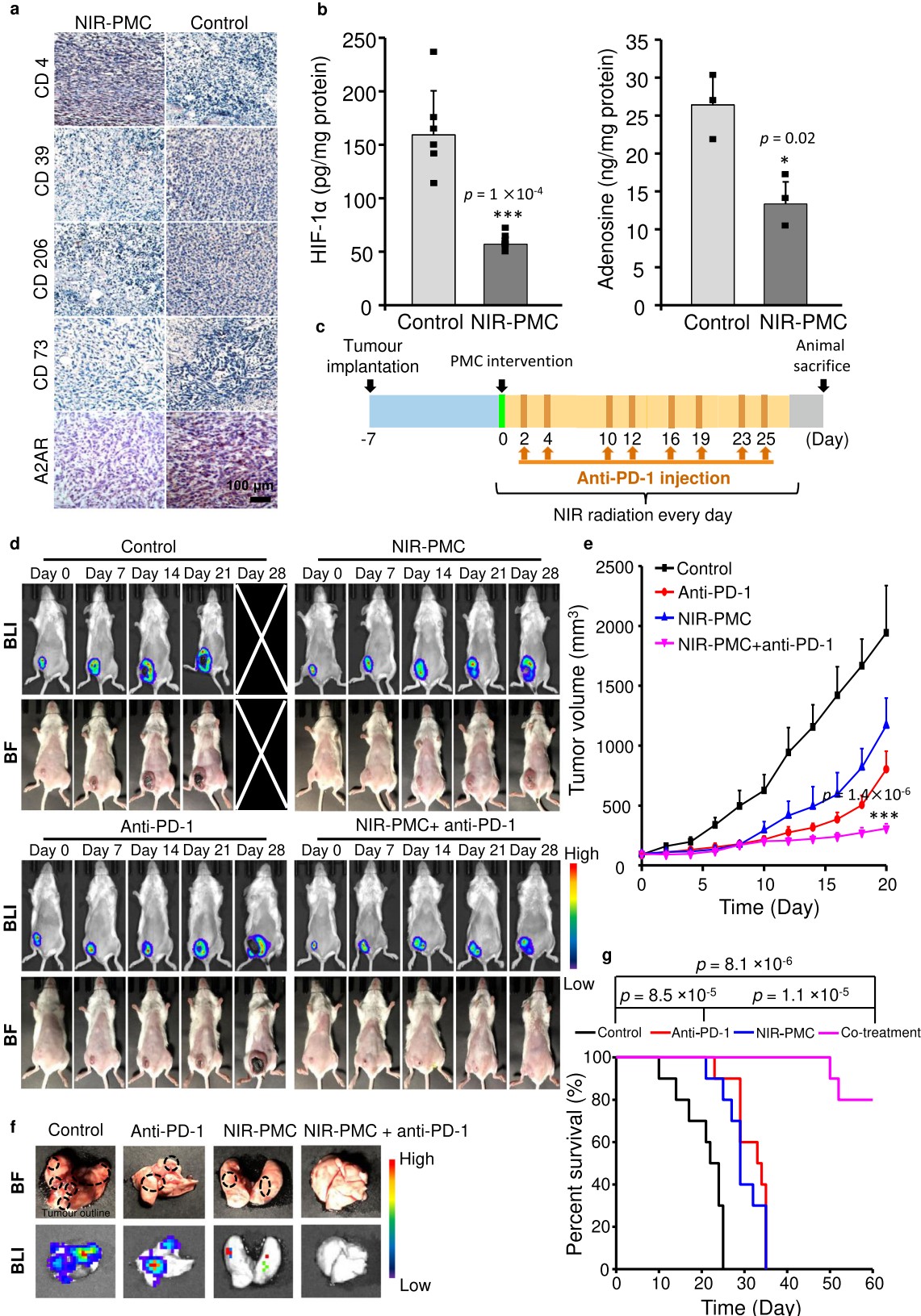

USA). Poly-L-Lysine (MW: 15000–30000, P4832), BMS-345541 (Cat. 401480), PMA (P1585) and Anti-β-actin antibody (A2066) were purchased from Sigma-Aldrich (St. Louis, Mo, USA). Anti-PD-1 was purchased from BioXCell (BE0146, Lebanon, USA). MTS assay kits were purchased from Promega (Madison, WI, USA). Penicillin, streptomycin, trypsin-EDTA and Dulbecco's modified Eagle medium (DMEM) and RPMI-1640 medium were purchased from Hyclone Laboratory (South Logan, Utah, USA). Anti-AKT (bsm-33278M) was purchased from Bioss (Beijing, China). Anti-PI3K (AF6241), anti-phospho-PI3K (AF3242) were purchased from Affinity Biosciences (USA). Insulin, Hoechst 33342, RIPA lysate, DCF-H, SYBR Green assay kit, LIVE/DEAD staining kit and Click-iT®EdU-594 assay kit were purchased from Thermo Fisher

**Fig. 6 | Therapeutic effects of NIR-PMCs combined with anti-PD-1 in breast cancer mice. a** Representative images of immunohistochemical staining (CD4, CD39, CD206, CD73 and A2AR) of breast tumours. **b** HIF-1α and adenosine productions in orthotopic breast tumours. HIF-1α or adenosine in tumour lysates were detected by ELISA and LC-MS, respectively ($n = 6$ and 3). The data are presented as the mean ± SD. *$p < 0.05$, ***$p < 0.001$ compared to untreated breast cancer mice by two-tailed Student's $t$-test. **c** Schematic illustration of combined treatment with NIR-PMCs and anti-PD-1. At 7 day post-tumour implantation (tumour size ~100 mm³), PMCs (25 μL, $3.6 \times 10^4$/mL) were intratumourally injected into tumour-bearing mice, followed by the mice being subjected to 300 mW/cm² NIR radiation for 28 days. A single injection of PMCs was administered throughout the tumour treatment procedure. Each mouse received 10 mg/kg anti-PD-1 twice a week. **d** Bioluminescence image (BLI) and bright field images (BF) of breast cancer-bearing mice. Animals that received different therapeutic agents were subjected to bioluminescence imaging by camera and an IVIS imaging spectrum system at 0, 7, 14, 21, and 28 days. **e**, Average tumour growth curves of mice. The tumour volume of individual mice was measured by a Vernier calliper every 2 days for 20 days ($n = 6$). The data are presented as the mean ± SD. ***$p < 0.001$ compared to untreated breast cancer mice according to two-tailed Student's $t$-test. **f** Ex vivo imaging of pulmonary metastasis. The animals that received different treatments were sacrificed at 21 days for visualization of metastatic nodules in the lungs. The nodules of pulmonary metastases are described by black outline. **g** Kaplan–Meier curves showing the survival rates of the indicated mice at 60 days ($n = 10$). The data are presented as ***$p < 0.001$, compared to the data for untreated hepatocarcinoma tumour-bearing rabbits according to Log rank test by SPSS statistics 17.0 software. Source data are provided as a Source data file.

Scientific (Grand Island, NY, USA). TNF-α recombinant protein (ab9642), EGF (ab9697), anti-phospho-AKT (ab38449), anti-mTOR (ab134903), anti-phospho-mTOR (ab109268), anti-ERK (ab32537) and anti-phospho-ERK (ab214036) was purchased from Abcam (Shanghai, China). 5-FU (F100149) were purchased from Aladdin (Shanghai, China). HIF-1 alpha staining kit was purchased from R&D (Minnesota, USA). Golgi inhibitor (Cat. 554724), Cytofix/Cytoperm buffer (Cat. 554722), Perm/Wash buffer (Cat. 554723) and anti-IFN-γ-FITC (Cat. 554411) were purchased from BD (Biosciences, Bedford, MA). Anti-mouse CD28 (Cat. 102102) antibody and anti-mouse CD3 antibody (Cat. 100302) were purchased from Biolegend (Shanghai, China). Hypoxyprobe™-1 Plus Kit was purchased from HPI Inc. (Burlington, MA, USA). Annexin V-FITC/PI apoptotic cell deaths assay kit (abs50001) were Absin Bioscience Inc (Shanghai, China). IL-15 (CLY200-07AF) was purchased from Cedarlane (Canada). Xylazine hydrochloride was purchased from Shengda Animal Products Co., Ltd (Jilin, China). PC membrane Transwell-24 (Cat. 3422) and Matrigel Matrix was purchased from Corning (NewYork, USA). Blue Green Algal cell growth medium (BG-11) was acquired from the Institute of Hydrobiology (Wuhan, China). Sodium alginate (molecular weight, 460 kDa; molar ratio of mannuronic acid to guluronic acid, 2/1) was purchased from Qingdao Bright Moon Seaweed Group Co., LTD (Qingdao, China). Upconversion nanoparticles (NaYREF$_4$, RE: Yb, Er@NaYF$_4$) were synthesized according to a reported method[75]. All other reagents were analytical grade.

## Algal microbes evolution test
Natural *Synechocystis sp*. 6803 (catalogue No. FACHB-898), *Synechococcus elongates* 7942 (catalogue No. FACHB-805), *Scenedesmus obliquus* (catalogue No. FACHB-417) and *Chlorella ellipsoidea* (catalogue No. FACHB-40) acquired from the Institute of Hydrobiology (Wuhan, China) were subjected to a selective evolution for acclimation of physiological conditions by stepwise alternations of growth conditions including temperature (25–37 °C) and medium composition (BG11 to DMEM). The culture temperature and media compositions were gradually changed to allow the evolution of microbes for survival in physiological conditions[76–78]. In detail, 3 mL algal microbes at stationary phase ($1.0 \times 10^8$ cells/mL) were dispersed in 10 mL BG11 media in baffled bottom flasks with vented caps (catalogue No. 4116-0125, Thermo Fisher Scientific Inc.) in a shaker for stepwise change of culture condition. Algal microbes cultured at 25 °C were used as control. The cell proliferation was calculated using Eq. (1) by detecting the optical density value at 660 nm (OD$_{660}$):

$$\text{Cell proliferation} \% = \frac{V_{\text{AT}} - V_{\text{mT}}}{V_{\text{A}} - V_{\text{M}}} \times 100\% \qquad (1)$$

Where $V_{\text{AT}}$ and $V_{\text{mT}}$ are the OD$_{660}$ values of algal microbes and culture media, respectively, at tested conditions; $V_{\text{A}}$ and $V_{\text{M}}$ are the OD$_{660}$ values of algal microbes and BG11 medium, respectively, at normal culture condition (25 °C).

Once the cells adapted to the tested temperature and showed >85% cell proliferation, we increased the culture temperature (1 °C). Otherwise, we sustained the temperature for another 24 h. After 30 days culture, the evolved algal microbes were acquired and preserved in BG11 media at 37 °C for subsequent tests. Next, we repeated the above procedure by stepwise changing medium composition from BG11 to DMEM. The evolved *Synechocystis sp*. 6803 cultured at 37 °C in DMEM was denoted as e-*S. sp*. 6803.

## PMC construction
e-*S. sp*. 6803 was collected by centrifugation at $1000 \times g$ for 10 min (Centrifuge 5810 R, Eppendorf) in 50 mL tubes and resuspended in PBS for PMC construction. The suspended cells were mixed with UCNPs at desired concentrations in 1.5 wt% sodium alginate solutions. After vortex, the mixture was extruded through a 0.5 mm needle into 0.1 M CaCl$_2$ solution (gelling bath) with an electrostatic droplet generator to prepare alginate-Ca beads. The beads were soaked in 0.05 wt% poly-lysine solution (volume ratio at 1:10) for 10 min to form polylysine coating. The coated microcapsules were suspended in 5 mL PBS and counted under a microscope. All fresh-made PMCs were cultured in the illuminating incubator (HerryTech KE-200, Shanghai, China) at 37 °C for further uses.

## Characterization of PMCs
The morphology and size of PMCs were determined by microscopy (FV1200, Olympus, Tokyo, Japan). Empty MCs, e-*S. sp*. 6803@MCs and PMCs at 3600/mL were added in an eight-well chamber for assessment of photoluminescence activities by upconversion confocal microscope with a 20× objective at 980 nm laser excitation. Photoluminescence spectra of UCNPs were measured with an Edingbour NanoSpectralyzer fluorimetric analyser (Applied Nano-Fluorescence, FLS980). To acquire oxygen release profile, freshly made PMCs at 3600/mL were suspended in 10 mL oxygen-free water that was subjected to 1 h vacuum extraction ($1.45 \times 10^{-3}$ psi) with reflux of CO$_2$. The PMC suspensions (10 mL) in transparent Eppendorf tubes were exposed to 980 nm radiation at 0, 100, 300, 900 mW/cm² for 0–60 min. A portable dissolved oxygen metre (JPB-70A oxygen sensor, Qiwei Instrument Co., Ltd. Hangzhou, China) was applied to record the dissolved oxygen concentrations. Suspensions containing UCNP@MCs were also included as blank.

## Cell culture
HCT116, HepG2, VX2, MCF-7, PANC-1, MGC-803, 4T-1, HeLa and A549 cells were purchased from ATCC (USA). STR analysis was performed to authenticate HeLa cell in our study. STR loci are amplified using fluorescently labelled PCR primers that flank the hypervariable regions. The result showed HeLa cell used in our study was matched with HeLa in ATCC (Supplementary Table 6). Yac-1 cells were purchased from National Collection of Authenticated Cell Caltures (Shanghai, China). MEF (mouse embryonic fibroblasts) cells were donated by Dr. Sudan He (Suzhou Institute of Systems Medicine, Chinese Academy of Medical

Science, China). Primary heart and kidney cells were acquired from BALB/c mice. In detail, dissected mouse heart and kidney were cut into small pieces (<3 mm), sufficiently washed by cold PBS (4 °C) to remove blood cells and transferred into dissociation tubes containing 5 mL working solutions of heart or kidney dissociation kits (Bio-leader Inc, Jiangxi, China). The tubes were fixed on the sleeve of a tissue dissociator (Bio-leader Inc, Jiangxi, China) for 1 min grinding at 100 g (30 s on, 30 s off), followed by 60 min enzymatic digestion at 37 °C. The cell suspensions were collected by filtration (70 μm). The heart and kidney cell pellets were counted and resuspended in cell culture media. All cell lines were cultured in DMEM medium or RPMI-1640 medium supplemented with 10% foetal bovine serum (Gemini, Woodland, USA), 100 U/mL penicillin and 100 μg/mL streptomycin at 37 °C 5% $CO_2$ or hypoxic condition (2% $O_2$ and 5% $CO_2$).

## Cell viability test
The tested cells were seeded in 96 well plates at $5 \times 10^3$/well for overnight incubation, followed by the addition of 25 μL PBS, empty MCs, UCNP@MCs and PMCs (25 μL, $3.6 \times 10^4$/mL). The cells and PMCs were cultured in a humidified hypoxic atmosphere with 2% $O_2$ and 5% $CO_2$ at 37 °C for 24 h with or without 300 mW/cm² NIR radiation for three intervals. After removal of the supernatants and residual PMCs, aliquots of 120 μL diluted MTS working solution were added to each well and incubated at 37 °C for an additional 2 h. The supernatants were transferred into new plates (100 μL/well) for absorbance detection at OD 490 nm by a Microplate Reader (Synergy NEO HTS, Biotek, USA). The cell viability was determined by Eq. 2:

$$\text{Cell viability } \% = \frac{(A_N - A_B)}{(A_C - A_B)} \times 100\% \qquad (2)$$

where $A_N$, $A_C$ and $A_B$ represent the absorbance of MTS substrate at 490 nm in treated, untreated cells and blank samples, respectively.

## Impacts of hyperoxia on the cytolytic activity of NK cells
The sorted NK cells ($2 \times 10^5$ cells/well) were co-cultured with fLuc-Yac-1 (mouse lymphoma) cells at the ratios of 1:1 or 2:1 in the presence or absence of PMCs, followed by three intervals of NIR exposure. After incubation for 6 h, the cell mixtures were collected to lyse Yac-1 cells by 100 μL Glo lysis buffer (Cat. E266A, Promega, WI, USA). After that, 50 μL/well One-Lite® Luciferase Assay System (Cat. DD1203-01, Vazyme, Nanjing, China) were added to measure the luminescence of cell lysates in a Microplate reader. The luminescence of Yac-1 cells was measured to determine cell death percentages that were used to assess the cytolytic activity of NK cells[79]. The percentage of cell death was determined by Eq. 3:

$$\text{Cell death } \% = \frac{I_0 - I_{\text{Lysate}}}{I_0} \times 100\% \qquad (3)$$

where $I_o$ and $I_{\text{lysate}}$ represent the luminescence intensity of cell lysate in untreated cells and co-cultured with NK cells, respectively.

## Construction of cancer cell spheroids
Matrigel Matrix (1.5 mL) was thawed in ice-bath at 4 °C and mixed with 0.5 mL HepG2 or MCF-7 cell suspensions ($4 \times 10^6$ cells/mL in pre-cooling DMEM). The solutions were gently mixed to avoid the generation of air bubbles. Aliquots of 300 μL suspensions were added into pre-cooling 24-well plates or 35 mm confocal dishes. The plates or dishes were cultured at 37 °C for 30 min to polymerize the matrix, followed by addition of 1 mL DMEM on the top of gel in each well/dish for further culture. Mature cancer cell spheroids were acquired after 7–10 day culture once the sizes of cell agglomerates are >50 μm.

## Lysis of cancer cell spheroids and tumour samples
The treated spheroids in 24-well plates were subjected to liquefying on ice by the cell recover solution (BD Biosciences, Bedford, MA). Cell spheroids were collected by centrifugation at $500 \times g$ for 5 min, washed by PBS and lyzed in PBS (50 μL) by three repeated freeze-thaw cycles in liquid nitrogen. Fresh tumours were weighted, and 50 mg samples were dissociated in 100 μL PBS at 4 °C by a homogenizer (OSE-Y30, TIANGEN, China). The homogenized tissue solutions and cell lysates were centrifuged at $500 \times g$ for 5 min to remove debris. The protein concentrations in supernatants were detected by Bradford assay. The lysates were stored at −20 °C for adenosine or cytokine analysis.

## NIR radiation treatment
A laser equipment (Xi'an Lei Ze Electronics Tech Co., Ltd, Shanxi, China) was used to provide NIR radiation for driving photosynthesis in PMCs by 980 nm laser at 0–900 mW/cm². PMCs in 10 mL media (PBS, BG11 or DMEM) were placed in transparent Eppendorf tubes and exposed to 100–900 mW/cm² NIR laser for 0–60 min. Cells and spheroids grown in multi-well plates or confocal dishes were exposed to 300 mW/cm² NIR radiation for three intervals (20 min for each interval, 15 min On and 5 min Off). Tumour-bearing mice and rabbits were exposed to three intervals (20 min for each interval, 15 min On and 5 min Off) of 300 mW/cm² and 900 mW/cm² NIR radiation each day, respectively.

## Confocal microscopy imaging
To visualize oxygen generation in PMCs, DCFH staining kit was used to visualize oxygen generation. PMCs, e-S. sp. 6803@MCs and UCNP@MCs at 1500/well were cultured in 1 mL DMEM media in 35 mm confocal dish for 12 h in the dark at 37 °C. Then the treated PMCs were incubated with 10 μg/mL DCFH for 10 min and then exposed to 300 mW/cm² 980 nm NIR radiation for 0, 1, 3 and 10 min. The fluorescence of DCF was immediately visualized by confocal microscopy at 488 nm excitation.

To assess the hypoxic status, mature HepG2 and MCF-7 cell spheroids in 35 mm confocal dishes were cultured with 1 mL fresh DMEM with or without PMC suspensions (4500/mL in DMEM). Then the spheroids were exposed to 300 mW/cm² NIR radiation for three intervals. Afterwards, spheroids were incubated with 100 μM Pimonidazole Hydrochloride for 12 h at 37 °C, washed by PBS and fixed by 4% paraformaldehyde for 30 min at room temperature. The cells in spheroids were permeabilized by 0.5% Triton X-100 in PBS for 1 h, and stained by 0.5 μg/mL FITC conjugated to mouse IgG1 monoclonal antibody (FITC-MAb1) in PBS for 6 h and 10 μg/mL Hoechst 33342 in PBS at room temperature for 30 min. The stained cells were washed by PBS thrice for microscopy imaging at excitation wavelengths of 405 nm and 488 nm.

For new daughter cell imaging, HepG2 and MCF-7 cells pretreated with or without 4500/mL PMCs were exposed to three intervals of NIR radiation. The treated cells were incubated with 1 ng/mL TNF-α for 12 h, or 40 μM BMS for 12 h. After that, the supernatants were replaced by 500 μL EdU working solution (10 μM) in a Click-iT®EdU-594 assay kit. After additional 2 h culture at 37 °C, the cells were fixed by 4% formaldehyde for 30 min at room temperature and permeabilized in 0.5% Triton X-100 in PBS for 20 min. After removal of the supernatants, and washing three times by 3% BSA, the click reaction cocktail containing 860 μL reaction buffer, 40 μL CuSO₄, 2 μL Azide 594 and 100 μL click additive solution was freshly prepared and added into the permeabilized cells. Following 30 min reaction at room temperature, the reaction cocktail was removed. The cells were washed twice by 3% BSA in PBS and incubated with 0.5 mL Hoechst 33342 (10 μg/mL) for nucleus staining. After 10 min incubation, the labelled cell samples were visualized by confocal microscopy (FV1200, Olympus, Japan)

## Western-blot assay

HepG2 cells were seeded into 6-well plates at a density of $6 \times 10^5$ cells/well. After overnight incubation, the cells were exposed to three interval NIR radiations in the presence or absence of $1.2 \times 10^4$/mL PMCs, followed by incubations with TNF-a (200 ng/mL), EGF (200 ng/mL) and Insulin (100 nM) for 10 min. After that, the supernatants were replaced with fresh media or media containing inhibitors for further incubations, including 20 μM BMS for 2 h, 2 mM 3-methyladenine for 1 h, 4 μM MK-2206 for 4 h, 100 nM rapamycin for 4 h or 100 μM PD98059 for 2 h. After sufficient washing by PBS, cell pellets were gathered and suspended in lysis buffer (10% glycerol, 20 mM Tris-HCl, pH 7.4, 150 mM NaCl, 1 mM $Na_3VO_4$, 1% Triton X-100, 25 mM β-glycerol-phosphate, 0.1 mM Phenylmethanesulfonyl fluoride) with 2% cocktail phosphatase (Cat: P5726, Sigma) and 5% protease (Cat: P5147, Sigma) inhibitors. The cell suspensions were vortexed for 10 s, and thereafter incubated on the ice for 30 min, followed by centrifugation at $2000 \times g$ for 20 min. The protein concentrations in supernatants were determined and adjusted by Bradford assay. The samples were separated on 8–15% SDS-PAGE gel (Beyotime, Shanghai, China) using tris-glycine-SDS buffer as running buffer in the Mini-PROTEAN Tetra System (Bio-Rad, CA, USA) at 100 V and transferred to a nitrocellulose membrane at 300 mA. The membranes were blocked with 5% milk (Biofrox, Einhausen, Germany) in 0.1% Tween 20/PBS at room temperature for 2 h, and then incubated with IκBα (1:1000), phosphorylated IκBα (1:1000), PI3K (1: 500), phosphorylated PI3K (1:500), AKT (1:1000), phosphorylated AKT (1:1000), mTOR (1:10000), phosphorylated mTOR (1:10000), ERK (1:1000), phosphorylated ERK (1:1000) and β-actin (1:5000) antibodies overnight at 4 °C. After thrice washing with 0.1% v/v Tween 20 in TBS (washing buffer) and incubation with HRP-conjugated secondary antibody (1:5000) for 2 h at room temperature, membranes were sufficiently washed. The freshly prepared ECL hypersensitive chemiluminescence solution was used to image the membranes by a multi-colour fluorescence chemiluminescence imaging analysis system (FluorChem M, Alpha, USA).

## Adenosine analysis by LC-MS

Spheroid (50 μL) and tumour (100 μL) sample lysates consisting of 0.1–0.4 mg proteins were mixed with 1 mL pre-cooled methanol and incubated at -20 °C for 16 h. The mixture was centrifuged at $20,000 \times g$ for 10 min at 4 °C to remove the protein precipitates. The metabolites in supernatants were dried with $N_2$ under a pressured gas blowing concentrator (VSD150-2, Woxin Instrument Manufacturing Co., China), and redissolved in 150 μL buffer A (0.1% formic acid in DI water). Adenosine standards dissolved in buffer A at 0–1000 nM were also prepared. Aliquots of 5 μL samples were injected and separated in a C18 reversed-phase column (2.1 mm × 150 mm, 100 Å, 3.5 μm). The LC-MS analysis was performed on an Agilent 1100 Series LC system (Agilent Technologies, Santa Clara, CA, USA) coupled with an LTQXL mass spectrometer (Thermo Fisher Scientific, Waltham, MA, USA). The mobile phase was programmed by an isocratic system consisting of 70% buffer A and 30% buffer B (100% ACN). The flow rate was maintained at 300 μL/min. MRM MS detection was operated in positive ionization mode to collect data. MS detection was conducted at 3 kV source voltage and 350 °C capillary temperature. The sheath and aux gas flow rates were set at 55 arb and 15 arb, respectively. MS data was analyzed by Xcalibur software. The concentrations of adenosine in tested samples were determined by its standard curve.

## Flow cytometry analysis

To examine whether NIR-PMC treatment induced apoptotic cell deaths, HepG-2 cells ($5 \times 10^5$ cells/well) were seeded in six-well plates and cultured in DMEM for 16 h. Then the supernatants were replaced by fresh media or DMEM containing 500 μg/mL 5-FU or PMCs at 3600/well and

exposed to NIR for three intervals. After 24 h incubation, the cells were collected and stained by an Annexin V-FITC/PI staining kit according to the manufacture's protocol before flow cytometry analysis.

Intracellular staining assay was performed to examine the effect of NIR-PMC treatment on IFN-γ production of T cells. Mouse T cells pre-activated with CD3 and CD28 antibodies were resuspended in RPMI-1640 supplemented with 10% FBS, then seeded in the lower chambers of 24-transwell plates ($2 \times 10^6$ cells/well, 2 mL) with 50 ng/mL PMA and 1 μL/mL Golgi inhibitor. PMCs were added into the upper chamber (3600/well) with 8.0 μm pore size on PC membranes. Then the whole cultures were exposed to three interval NIR radiation. After 18 h, the cells were spun down at 700 g for 5 min, and the pelleted cells were fixed in BD Cytofix/Cytoperm buffer for 30 min. The fixed cells were washed by BD Perm/Wash buffer (Cat. 554723, BD Biosciences) once, and then the cells were resuspended in BD Perm/Wash buffer containing 5 μL/test anti-IFN-γ-FITC (Cat. 554411, BD Biosciences) for incubation at room temperature for 30 min. After thrice washing in BD Perm/Wash buffer, the resuspended cells were subjected to flow cytometry (FACSVerse, BD) analysis. Data analysis was carried out using FlowJo software (Tree Star Inc.)

## Animal treatment

Female BALB/c mice (6–8 weeks old, -20 g) were obtained from the Experimental Animal Center of the Gempharmatech Co. Ltd in Nanjing, China. Female New Zealand White rabbits (6 months old, weighing 2–2.5 Kg) were obtained from Qingdao Kangda Rabbit Co., Ltd. (Qingdao, China). All animals were grown in an animal facility under filtered air conditions (22–25 °C), 12 h dark/light cycle (8:00-20:00 light; 20:00-8:00 dark), and relative humidity (40–70%) in plastic cages with sterilized wood shavings for bedding. All animal experiments were strictly performed under the guidelines of the Chinese Council for Animal Care approved by the Animal Care Committee of the Laboratory Animals in Soochow University (No. ECSU-202109A0401). The approved humane endpoints were applied on the tumour-bearing animals once they meet one of the following criteria: (i) the tumour diameter is >2 cm for mice or the tumour volume is >80 cm³ for rabbits; (ii) the eating, drinking or movement of animals is severely affected. To develop the hepatic VX2 tumours in rabbits, VX2 cell suspensions ($2 \times 10^6$ cells, 200 μL) were implanted into the thigh muscles of donor rabbits. Once the tumour sizes were >2 cm (-2 weeks), the donor rabbits were anesthetized by intravenous injection at a lethal dose 2 mL/Kg of xylazine hydrochloride for the harvest of tumour tissues. Each tumour was minced into 1 mm³ piece by ophthalmic scissors under sterile conditions. The recipient rabbits were anesthetized by intramuscular injection of xylazine hydrochloride (250 μL/Kg). A minced tissue fragment was directly delivered percutaneously into the subcapsular parenchyma of the left hepatic lobe of the recipient rabbit by percutaneous puncture technique under a 16-slice CT spiral scan (Brilliance-16, Phillips, USA) guidance. The rabbits were housed and examined by CT imaging until the tumour volumes reached around 1 cm³. The hepatocarcinoma-bearing rabbits with similar tumour size were divided into two groups by throwing dice, including vehicle control ($n = 13$), NIR-PMC group ($n = 13$). The rabbits were anesthetized for a single intratumorally injection of PMC suspensions (500 μL, $3.6 \times 10^4$/mL) at 14 days. NIR radiations at 900 mW/cm² were exposed to animals for three intervals (20 min in each interval) each day. The tumour size was monitored by CT scanning (MHCT brilliance 16, Philips, Holland) every 2 weeks.

fLuc-4T1 cells donated by Dr. Zhimou Yang (Nankai University, Tianjin, China) were suspended in 100 μL PBS at $2 \times 10^8$ cells/mL and injected into the mammary fat pads of each animal. The tumour volumes were examined every 2 days and calculated by Eq. 4:

$$\text{Tumor volume (mm}^3) = width^2 \times \frac{length}{2} \qquad (4)$$

The 4T-1 cell inoculated mice were collected for further treatments once the tumour sizes reached to ~50 mm³. All the qualified animals were randomly divided into four groups by throwing dice for the following treatments, including intratumor injection of 25 μL saline (vehicle Ctrl), intratumor injection of $3.6 \times 10^4$/mL PMCs in 25 μL saline coupled with 300 mW/cm² NIR-radiation for three intervals (15 min in each interval) per day (NIR-PMC group), intravenous injection of 10 mg/Kg anti-PD-1 (100 μL, twice a week) (anti-PD-1 group), and cotreatment by NIR-PMC and anti-PD-1 (cotreatment group). Notably, the animals in PMC, NIR-PMC and cotreatment groups merely intratumorally injected with PMCs at the 7th day. For combined treatment of breast cancer by PMC implants and immunotherapy in mice, the detailed animal treatment procedure was illustrated in Fig. 5b. The combination index (CI) of hyperoxic (300 mW/cm² NIR for three interval) and 10 mg/Kg anti-PD-1 (100 μL, twice a week) was calculated according to a reported formula:

$$CI = \frac{AB}{A \times B} \tag{5}$$

Where AB, A and B are the tumour volumes in NIR-PMC, anti-PD-1 and co-treatments, respectively. CI > 1 indicates antagonism, CI = 1 indicates additivity, and CI < 1 indicates synergy. The tumours were imaged by IVIS imaging spectrum system (PerkinElmer, ME, USA) and Canon camera (Japan). The mice were fully anesthetized by an overdose of sodium pentobarbital (400 mg/kg) and sacrificed to collect tumours and lungs. The tissue samples were stored in liquid nitrogen for cytokine and adenosine measurements or fixed for H&E staining or immunostaining of A2AR, CD4, CD39, CD206 and CD73 expression.

## Detection of the partial pressures of oxygen in tumours by Clark electrode
Breast cancer bearing mice with tumour volumes at ~50 mm³ were randomly divided into five groups by throwing dice for the following treatments, including intratumorally injection of 25 μL saline (vehicle Ctrl), receiving 60% hyperbaric oxygen for 1 h (Hyperbaric $O_2$ group), intratumorally injection of $3.6 \times 10^4$/mL PMCs in 25 μL saline (PMC group), intratumorally injection of $3.6 \times 10^4$/mL empty microspheres in 25 μL saline coupled with 300 mW/cm² NIR exposure each day (NIR group), and intratumorally injection of $3.6 \times 10^4$/mL PMCs in 25 μL saline coupled with 300 mW/cm² NIR exposure each day (NIR-PMC group). Three animals were included in each group. The partial pressures of oxygen ($pO_2$) in tumours were detected at different depths by a Clark sensor (0.4 mm in diameter) on a $pO_2$ monitor (POG-203, Unique Medical, Tokyo, Japan) at 110 V according to the manufacturer's protocol.

## Isolation and expansion of murine T and NK cells
For preparation of splenocytes, BALB/c mice (age: 6–8 weeks; weight: 20 g) were sacrificed by $CO_2$ inhalation to collect spleens. The spleens were ground in 10 mL 1640 RPMI media (10% foetal bovine serum, 100 U/mL penicillin, 100 μg/mL streptomycin). The mixture was filtered by 200 mesh nylon membrane (Cat. 7061011, Dakewe, China) to remove tissue debris. The suspensions were centrifuged (Allegra 64 R, Beckman) at $500 \times g$ for 5 min to collect cell pellets, followed by twice washing in PBS. The cell pellets were dispersed and incubated in 5 mL 1× RBC red cell lysis buffer (8.26 g/L $NH_4Cl$, 1 g/L $KHCO_3$ and 0.037 g/L EDTA in DI $H_2O$) at room temperature for 5 min. Subsequently, RPMI-1640 media (5 mL) were added to stop the lysis reaction and centrifuged to collect the splenocytes, which were used for expansion of T cells and NK cells as described below.

For expansion of T cells, the anti-mouse CD28 and CD3 activation antibodies were dissolved in PBS at CD3 and were added into 6-well plates (1.5 mL/well) to treat well surfaces at 4 °C for 12 h. Afterwards, antibody solutions in each well were discarded and the coated plates were ready for expansion of T cells form mouse splenocytes. The splenocytes which were prepared as described above were resuspended in RPMI-1640 media at $2 \times 10^6$/mL, added into the CD28/CD3 antibodies-coated plates (5 mL/well) and incubated for 48 h in the presence of 200 IU/mL IL-2 for expansion of T cell. Then the pre-activated T cells were collected and transferred into 24-well plates ($2 \times 10^6$/mL, 2 mL/well) for further test of IFN-γ production.

For NK cell expansion, the splenocytes ($2 \times 10^6$/mL, 5 mL/well) were added in six-well plates in the presence of 20 ng/ml mouse IL-15 and 1000 IU/ml mouse IL-2. The cell media were changed every 2 days. After 7 days, the resultant cells were labelled with 1 μL/1 × 10⁶ cells anti-CD3-APC (Cat. 30041, Biolegend) and 1 μL/1 × 10⁶ cells anti-NK1.1-PE (Cat. 557391, BD Biosciences) for 30 min. The stained cells were collected to sort out the NK1.1 + CD3- NK cells using flow cell sorter (FACSMelody, BD). The sorted NK cells were transferred to 24-well plates for further test of cytolytic activity.

## Real-time quantitative PCR analysis
Rabbit tumours (20–30 mg) were added into slurry pipes with 1 mL Trizol to dissociate the tumour tissues with a homogenizer (PRO200, FLUKO, China) for 2 min and then placed on ice for 15 min to allow cell lysis. Afterwards, the supernatants were obtained by centrifugation at 4 °C, $6000 \times g$ for 15 min. Then, 0.5 mL supernatant was mixed with chloroform (0.2 mL) and oscillated for 2 min to extract nucleic acid. After stewing for 5 min on ice, the mixture was centrifuged at 4 °C $6000 \times g$ for 15 min. The upper aqueous solution (0.2 mL) was collected and transferred into a new RNase-free Eppendorf tube. Isopropanol (0.2 mL) was added to the supernatants and centrifuged at 4 °C, $6000 \times g$ for 15 min. The pellets were collected and washed by 75% ethanol. The resulting RNA was redissolved in 20 μL diethyl pyrocarbonate (DEPC) water to quantify the concentration by Nanodrop 2000c Spectrophotometer (Thermo Fisher, USA) at OD 260 nm. The PCR analysis was performed in Anhui Leaobei Biotechnology Co. LTD (Tongling, China) by a SYBR Green assay kit (Thermo Fisher, USA) under a Real-Time PCR System (ABI-7300, USA). The data were analyzed by ABI Prism 7300 SDS Software and the mRNA levels of each gene in two groups ($n = 3$) were normalized to that of GAPDH. The primers used are listed in Supplementary Table 2.

## In vivo biosafety assessment
Healthy BALB/c mice (age: 6–8 weeks; weight: 20 g) were randomized into three groups by throwing dice. The animals were anesthetized for administrations of PMC suspensions (25 μL, $3.6 \times 10^4$/mL) or equal volume of PBS on 0 day by intraperitoneal or subcutaneous injection. NIR radiations at 300 mW/cm² were exposed to animals each day. The animals' behaviour and appearance were inspected periodically after the injection. The treated mice were sacrificed by inhalation of $CO_2$ on 1, 3 and 8 days to collect organs (heart, liver, lung, kidney, brain, spleen and skin) and bloods. The organ tissues were fixed for H&E or TUNEL staining. The blood samples (~400 μL each mouse) were subjected to detection of routine blood indicators by Mindray BC-2800Vet haematology analyser (Mindray Global, China).

## Statistics and reproducibility
All experiments were repeated at least thrice with three to ten replicates. Data were expressed as mean ± standard deviation (SD) from at least three replicates. All confocal imaging and immunohistochemical staining imaging were repeated at least three replicates. Data analysis was performed by two-tailed Student's $t$-test or Log rank test via SPSS statistics 17.0 software. The difference was regarded as statistical significance if $p < 0.05$.

## Reporting summary
Further information on research design is available in the Nature Research Reporting Summary linked to this article.

## Data availability

Source data are provided with this paper. The remaining data are available within the Article, Supplementary Information or Source Data file. The imaging data are deposited on a public repository, Zenodo Dataverse, by following https://doi.org/10.5281/zenodo.6588765 Source data are provided with this paper.

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

## Acknowledgements

This work was supported by the grants from the National Natural Science Foundation of China (21976126 to R.L.), the Natural Science Foundation of Jiangsu Province (BK20211545 to R.L.), National Key R&D Programme of China, Ministry of Science and Technology of China (2020YFA0710700 to R.L.).

## Author contributions

R.L. conceived the idea and designed the experiments. W.W. performed most experiments. H.Z. and Ju.J. constructed the PMCs. H.Z. performed cell proliferation, HIF-1α, qPCR, new daughter cell imaging, and adenosine production test. Z.L. performed rabbit tumour implantation experiments. Ju.J. performed mice tumour implantation experiments. D.J., X.S., and J.C. contributed in T and NK cell extraction as well as activity test. H.W. and Ji.J. tested oxygen level in tumour. X.C., Q.X. quantified the HIF-1α level in cells and tumours. M.G. contributed in cell viability test. T.X. contributed the idea of immunotherapy in the hyperoxia microenvironment. The writing of the manuscript was led by R.L. with participation from W.W.

## Competing interests

The authors declare no competing interests
