## [Peer Review File · Nature Communications]

Title: Engineering Micro Oxygen Factories to Slow Tumour Progression via Hyperoxic MicroenvironmentsEditorial Note: This manuscript has been previously reviewed at another journal that is not operating a transparent peer review scheme. This document only contains reviewer comments and rebuttal letters for versions considered at *Nature Communications*.

REVIEWERS' COMMENTS

Reviewer #1 (Remarks to the Author):

The manuscript is suitable for publication with minor changes.

The authors should state that a limitation of the method is that 24 h after a 60 minute exposure to NIR-PMCs the tumors of mice treated with the PMCs return to back to hypoxic conditions = 10mm Hg (see Figure 1E). Even if the NIRs are delivered daily O₂ levels go up sharply and then decrease back to hypoxia. The effect of the cancer cell phenotype that is induced by O₂ cycling should be a consideration in follow-on studies. This should be mentioned as a caveat.

Reviewer #4 (Remarks to the Author):

The authors have well addressed my questions. Several important experiments were supplemented to support the conclusions. I believe the novelty and experimental validation of this paper meet the requirements of Nature Communications, so I recommend it for publication.

Reviewer #1 (Remarks to the Author):

The manuscript is suitable for publication with minor changes.

The authors should state that a limitation of the method is that 24 h after a 60 minute exposure to NIR-PMCs the tumors of mice treated with the PMCs return to back to hypoxic conditions = 10mm Hg (see Figure 1E). Even if the NIRs are delivered daily O₂ levels go up sharply and then decrease back to hypoxia. The effect of the cancer cell phenotype that is induced by O₂ cycling should be a consideration in follow-on studies. This should be mentioned as a caveat.

Response: We thank the reviewer for thinking that our work is suitable for publication. To accommodate the reviewer's concern, we have added a short discussion in lines 350-354 of the revised manuscript, as "Although the NIRs are applied o animals daily, the pO₂ in tumours reached peaks (27.2-35.4 mmHg) at 1 h post-NIR exposure and returned back to hypoxic status (10 mm Hg) in 24 h. The periodic oxygen supplementation may impact the metabolic network and phenotypes of cancer cells, which should be considered in follow-on studies."

Reviewer #4 (Remarks to the Author):

The authors have well addressed my questions. Several important experiments were supplemented to support the conclusions. I believe the novelty and experimental validation of this paper meet the requirements of Nature Communications, so I recommend it for publication.

Response: We thank the reviewer for thinking that our work is suitable for publication.